# The structure of phosphatidylinositol remodeling MBOAT7 reveals its catalytic mechanism and enables inhibitor identification

Kun Wang[1,2], Chia-Wei Lee[1,2], Xuewu Sui [1,2,9], Siyoung Kim[3], Shuhui Wang [2], Aidan B. Higgs[1,2], Aaron J. Baublis[1,2,4], Gregory A. Voth [5], Maofu Liao [2,6,11] ✉, Tobias C. Walther [1,2,4,7,8,10,11] ✉ & Robert V. Farese Jr [1,2,7,10,11] ✉

Cells remodel glycerophospholipid acyl chains via the Lands cycle to adjust membrane properties. Membrane-bound *O*-acyltransferase (MBOAT) 7 acylates lyso-phosphatidylinositol (lyso-PI) with arachidonyl-CoA. *MBOAT7* mutations cause brain developmental disorders, and reduced expression is linked to fatty liver disease. In contrast, increased *MBOAT7* expression is linked to hepatocellular and renal cancers. The mechanistic basis of MBOAT7 catalysis and substrate selectivity are unknown. Here, we report the structure and a model for the catalytic mechanism of human MBOAT7. Arachidonyl-CoA and lyso-PI access the catalytic center through a twisted tunnel from the cytosol and lumenal sides, respectively. N-terminal residues on the ER lumenal side determine phospholipid headgroup selectivity: swapping them between MBOATs 1, 5, and 7 converts enzyme specificity for different lyso-phospholipids. Finally, the MBOAT7 structure and virtual screening enabled identification of small-molecule inhibitors that may serve as lead compounds for pharmacologic development.

Phospholipids are synthesized in the endoplasmic reticulum (ER) via a series of enzymatic steps that sequentially add acyl chains to a glycerol backbone[1,2]. Glycerol acyltransferase enzymes[3] preferentially add a saturated fatty acid to the *sn*−1 position, and acylglycerol acyltransferase enzymes usually esterify an unsaturated fatty acid to the *sn*−2 position. To adjust membrane properties according to need and to store specific fatty acids as precursors for bioactive lipids, cells remodel the phospholipid acyl chains in enzymatic reactions known as the Lands cycle[4]. In this process, the *sn*−2 fatty acyl chain is removed by hydrolysis, and the lysophospholipid product is re-esterified with an acyl-CoA to form a phospholipid with a different acyl chain[4,5].

Re-acylation is carried out by lysophospholipid acyltransferase enzymes, including various membrane-bound *O*-acyltransferases

[1]Department of Molecular Metabolism, Harvard T.H. Chan School of Public Health, Boston, MA, USA. [2]Department of Cell Biology, Harvard Medical School, Boston, MA, USA. [3]Pritzker School of Molecular Engineering, University of Chicago, Chicago, IL, USA. [4]Harvard T.H. Chan Advanced Multi-Omics Platform, Harvard T.H. Chan School of Public Health, Boston, MA, USA. [5]Department of Chemistry, Chicago Center for Theoretical Chemistry, James Franck Institute, and Institute for Biophysical Dynamics, The University of Chicago, Chicago, IL, USA. [6]School of Life Sciences, Southern University of Science and Technology, Shenzhen, China. [7]Broad Institute of MIT and Harvard, Cambridge, MA, USA. [8]Howard Hughes Medical Institute, Boston, MA, USA. [9]Present address: Department of Biochemistry and Biophysics, College of Agriculture and Life Sciences, Texas A&M University, College Station, TX, USA. [10]Present address: Cell Biology Program, Sloan Kettering Institute, Memorial Sloan Kettering Cancer Center, New York, NY, USA. [11]These authors contributed equally: Maofu Liao, Tobias C. Walther, Robert V. Farese Jr. ✉e-mail: liaomf@sustech.edu.cn; twalther@mskcc.org; rfarese@mskcc.org

(MBOAT), such as MBOATs 1, 2, 5, and 7 in humans[5]. MBOAT1 and MBOAT5 remodel phosphatidylserine (PS) and phosphatidylcholine (PC), respectively[6,7]. MBOAT2 remodels phosphatidylethanolamine (PE) and phosphatidic acid (PA), and to a lesser extent PC and PS[7]. MBOAT7 uniquely remodels phosphatidylinositol (PI), catalyzing preferentially the esterification of arachidonyl CoA to lyso-PI[5,7] (Fig. 1a).

MBOATs form a large class of ER enzymes (11 in humans) that acylate small molecules, lipids, or proteins[8]. The structures of several MBOAT enzymes have been determined by cryo-electron microscopy (cryo-EM)[9–18]. Some MBOATs, such as porcupine and hedgehog acyltransferase, acylate polypeptides in the ER lumen[10,11,18]. Other MBOATs catalyze neutral lipid synthesis (e.g., DGAT1 and ACAT1/ACAT2) and

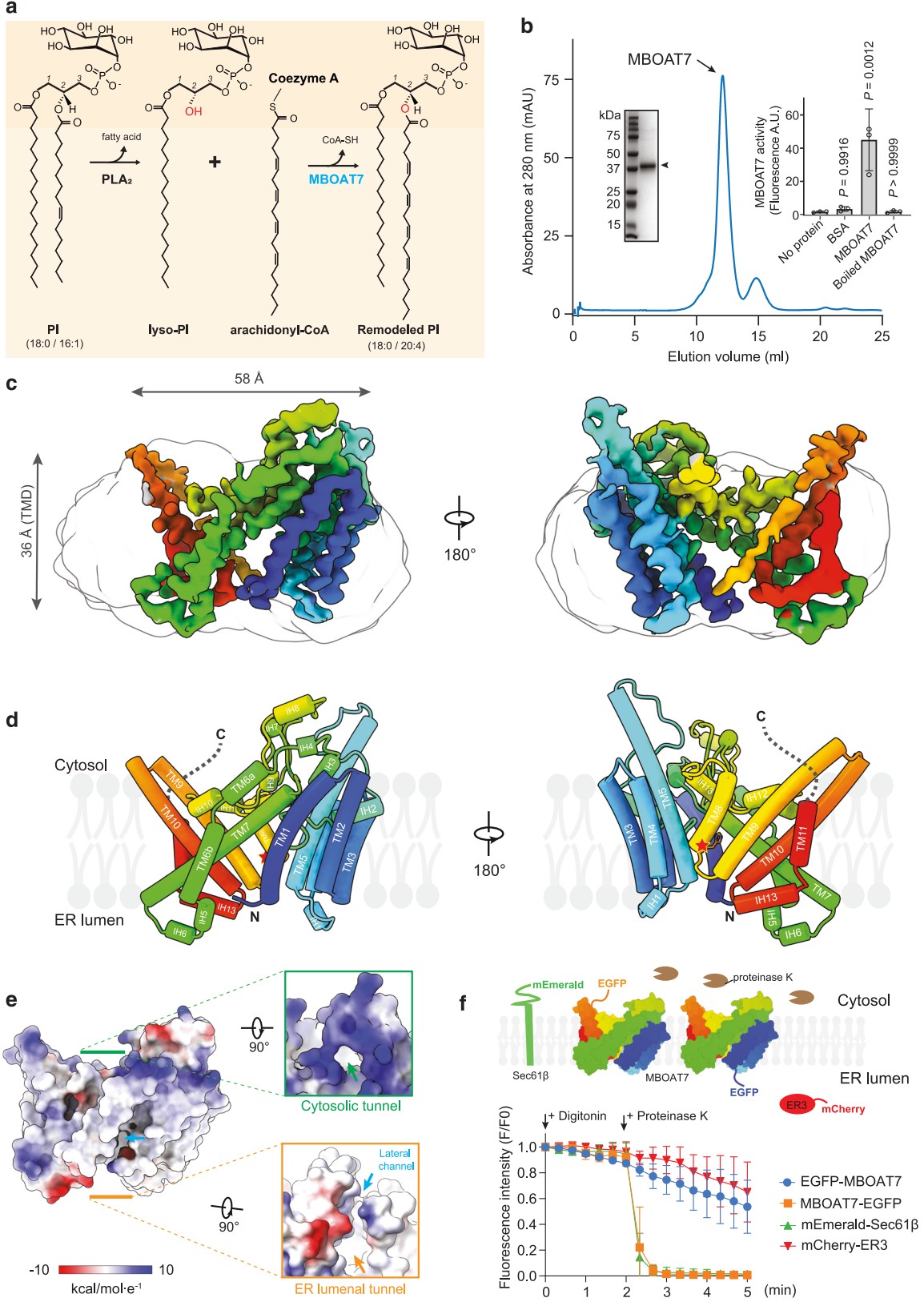

**Fig. 1 | Cryo-EM structure of human MBOAT7. a** Diagram depicting the Lands cycle and MBOAT7-mediated PI remodeling in the membrane. Newly synthesized PI contains mainly monounsaturated *sn-2* acyl chains. Phospholipase A2 (PLA$_2$) hydrolyzes the *sn-2* ester bond and produces lyso-PI. MBOAT7 specifically re-acylates lyso-PI with arachidonyl-CoA as the acyl donor to produce the remodeled PI molecules. The *sn-2* hydroxyl group is highlighted in red. **b** Size-exclusion profile of purified MBOAT7 reconstituted in PMAL-C8. Inset left, SDS-PAGE analysis of the MBOAT7 peak. The arrowhead denotes the MBOAT7 band, which migrates faster than its 53-kDa mass. Inset right, activity analysis of the purified MBOAT7 protein with the fluorescence monitoring of the production of the byproduct CoA-SH (mean ± standard deviation [SD], *n* = 3 independent experiments). Analysis was performed using one-way ANOVA with Dunnett's *post hoc* test. **c** Cryo-EM map of human MBOAT7. The amphipol micelle is shown as a transparent gray outline. Map is contoured at 6 σ. The map is rainbow-colored, corresponding the colors of secondary elements in the models in **d**. **d** Cylinder representation of the human MBOAT7 model. The red star indicates the catalytic residue His356. Dashed lines indicate the disordered C-terminal region. TM, transmembrane; IH, intervening helix. **e** Representation of MBOAT7 with an electrostatic surface. The inserts indicate the entries to the two tunnels from two sides of the membrane. Arrows indicate the specific features. **f** The topology of MBOAT7 in membranes. Top, Schematic depiction of the fluorescence protease protection (FPP) assay. Plasma membrane is permeabilized by low concentrations of digitonin without disrupting the ER membrane. Subsequent addition of proteinase K selectively cleaves the fluorescent proteins on the cytoplasmic side of the exposed ER membrane, leaving the fluorescent proteins exposed in the lumen intact. Bottom, the quantification of the fluorescence intensities of the whole time series of the FFP assay. The time points when digitonin or proteinase K were added were indicated (Mean ± standard error of the mean [SEM], *n* = 3 independent experiments). Source data are provided as a Source Data file.

act as multimers with a catalytic histidine residue that is buried deeply within the membrane[12–16]. For these MBOATs, the acyl-CoA substrate accesses the catalytic center through a tunnel that reaches from the cytoplasmic to the lumenal side of the ER. A lateral gate allows for entrance of the acyl acceptor substrate and exit of the neutral lipid product into the membrane.

There are fewer insights into the molecular mechanisms for MBOATs involved in the Lands cycle. The cryo-EM structure of chicken MBOAT5 (cMBOAT5) reveals a dimer, with each protomer having a fatty acyl–binding tunnel and a T-shaped chamber for esterification of arachidonyl-CoA with lyso-PC[9]. It remains unclear, however, how different MBOATs recognize different lysophospholipid substrates.

We focused our studies on MBOAT7 because of its unique function in PI metabolism and because understanding its catalytic mechanism has important implications for human physiology and disease. MBOAT7 deficiency in humans leads to developmental disorders of the central nervous system, with intellectual disability, epilepsy and autism spectrum disorder[19–21]. Deficiency of MBOAT7 function in liver is associated with increased lipogenesis in humans, likely by activating SREBP-1c and de novo lipogenesis[22], and is an important genetic risk factor for developing non-alcoholic fatty liver disease and other liver diseases[23–28]. Conversely, increased *MBOAT7* expression is correlated with detrimental outcomes in cancers, such as hepatocellular carcinoma and clear cell renal carcinomas[29,30], suggesting that inhibitors of MBOAT7 may be useful therapeutic agents. Indeed, deletion of *MBOAT7* inhibited tumor formation in clear cell carcinoma animal models[29].

Here we determined MBOAT7's molecular structure by single-particle cryo-EM. Combining structural insights with molecular dynamics (MD) simulations and biochemical data, our findings suggest a model for catalysis and substrate specificity of phospholipid remodeling MBOATs. The MBOAT7 structure further enabled us to identify, by virtual screening, small-molecule lead compounds that specifically inhibit MBOAT7.

## Results

### Cryo-EM study of human MBOAT7 reveals characteristic structural features

To enable biochemical and structural analyses of human MBOAT7, we purified recombinantly expressed enzyme in detergent and reconstituted it into amphipol PMAL-C8 (Fig. 1b and Supplementary Fig. 1a). Purified MBOAT7 was similarly active in detergent or PMAL-C8 (Supplementary Fig. 1b). MBOAT7 in PMAL-C8 exhibited PI synthesis activity with a Vmax of ~300 pmol/min/μg protein, and Kms for arachidonyl-CoA and lyso-PI of 46 μM and 26 μM, respectively, which are similar to those of other MBOATs[10,12,31] (Supplementary Fig. 1c, d). MBOAT7 was most active with arachidonyl-CoA (C20:4) as the acyl donor, followed by other unsaturated acyl-CoAs, and strongly preferred lyso-PI as the acyl acceptor (Supplementary Fig. 1e, f).

We used single-particle cryo-EM to obtain a density map of MBOAT7 at an overall resolution of ~3.7 Å (Fig. 1c, Supplementary Fig. 2), allowing us to build a model of the entire protein except the C-terminal region (442–472), which was not visible and presumably disordered (Fig. 1d, Supplementary Fig. 2). Overall, the structure of MBOAT7 was similar to the model predicted by AlphaFold[32,33] with root-mean-squared distance (RMSD) ~1 Å on the backbones and ~2.2 Å on the entire protein (residues 1–441, Supplementary Fig. 3). MBOAT7 comprises 11 transmembrane (TM) helical segments, connected by short intervening helices (IH) and loops. The monomeric protein occupies a space of ~36 × 58 × 28 Å, embedded in the membrane (Fig. 1c), resembling the shape of a saddle.

The fold of MBOAT7 is similar to that of other MBOATs[9,10,12,14,17,18]. Two TM bundles (TM1–5, TM9–11) tilt away from each other, generating space in the center for the catalytic chamber. The chamber between these bundles is sealed by two long, diagonal TM segments (TM6 and TM7) on one side and TM8 on the other. The diagonal helix TM6 is broken into two short helices (TM6a and TM6b) (Fig. 1d); in contrast, a corresponding helix in cMBOAT5 is continuous[9] (Supplementary Fig. 4e). TM8, containing the putative catalytic His356 residue, resides in the cavity between the two TM bundles. TM8 exhibits weaker density than the other TMs, indicating flexibility or multiple conformations in the apo state of the enzyme (Supplementary Fig. 2f, TM8 density shown in two σ values).

Besides the "MBOAT core" (TM3–10, Supplementary Fig. 4a), other regions of MBOAT7 are distinct from other MBOAT structures (Supplementary Fig. 4b and c). For example, TM1-2 have features that are different from DGAT enzymes. MBOAT7's N-terminal region contains only a short sequence before the first TM helix (TM1) and lacks sequences homologous to the longer N-terminal region of DGAT1, which mediates enzyme dimerization[12,13]. Also, DGAT1 enzymes have a large lateral gate that opens into the plane of the membrane, which presumably provides access for diacylglycerol substrates and an exit path for triacylglycerol products[12,13]. In contrast, TM helices 1 and 2 of MBOAT7 occlude the equivalent lateral gate region (Supplementary Fig. 4d).

A tunnel twists through MBOAT7, connecting the cytoplasmic and the ER lumenal sides of the enzyme, with the active site located in the middle of this tunnel (Fig. 1e). This suggests a model similar to those proposed for other MBOATs[9–11], in which acyl-CoA accesses the active site from the cytosolic side of the membrane and the acyl-acceptor, lyso-PI, enters the enzyme from the lumenal side. In MBOAT7, the lateral opening of the tunnel has an extended channel that branches off to the lumenal leaflet of the ER membrane.

Our model predicts that the N- and C-termini of MBOAT7 are on opposite sides of the ER membrane, with the N-terminus in the ER lumen and the C-terminus in the cytoplasm. To experimentally test this model, we performed fluorescence protease protection assays[34] with MBOAT7 fused to EGFP at either the N- or C-terminus (Fig. 1f). Controls

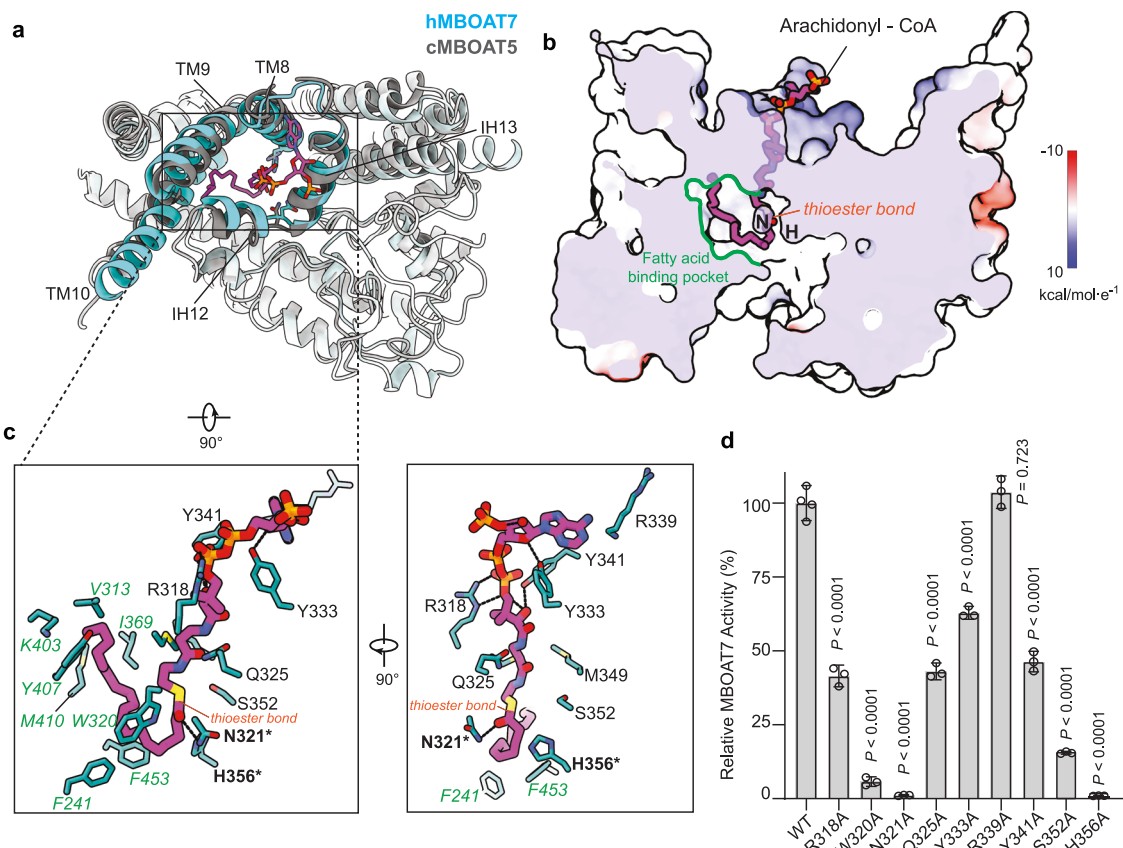

**Fig. 2 | Arachidonyl-CoA accesses the MBOAT7 catalytic chamber via a tunnel from the cytosolic leaflet of the membrane. a** Top view of the arachidonyl-CoA molecule docked into the MBOAT7 cytosolic channel. The human MBOAT7 (hMBOAT7) structure is superimposed with the chicken MBOAT5 structure complexed with arachidonyl-CoA (PDB 7F40). The key helices making up the chamber, including TM8–10 and IH12–13, are highlighted in cyan. The RMSD of 99 α-carbons within these regions between MBOAT5 and MBOAT7 is -1.15 Å. **b** A cutting-in view of the catalytic chamber and side pocket bound with arachidonyl-CoA. MBOAT7 is represented with electrostatic surface. The positions of catalytic residues H356 and N321 are indicated by H and N, respectively. A twisted tunnel spans the enzyme from the cytosolic leaflet to the ER lumenal leaflet. The cytosolic access to the tunnel has a side pocket that accommodates arachidonyl-CoA binding. **c** The interaction between arachidonyl-CoA and MBOAT7 residues. Polar interactions are shown with dashed lines. The catalytic residues are highlighted in bold with an asterisk. The residues that comprise the fatty acid binding pocket are colored in green. **d** Activity of the arachidonyl-CoA binding channel alanine mutations. The enzymatic activities of mutants were normalized as the percentage of that of the wild-type MBOAT7 (mean ± SD, $n = 3$ independent experiments except $n = 4$ for WT). Analysis was performed using one-way ANOVA with Dunnett's *post hoc* test. Source data are provided as a Source Data file.

included expressed mEmerald-Sec61β (fluorescent protein cytosol-exposed) and mCherry-ER3 (a lumenal protein). We used low concentrations of digitonin to selectively permeabilize the plasma membrane, but not the ER membranes. Upon proteinase K treatment, the fluorescent signal of mEmerald-Sec61β was rapidly attenuated, whereas the signal of mCherry-ER3 was protected (Fig. 1f). For MBOAT7, with proteinase K treatment, signal was lost for the C-terminally, but not the N-terminally tagged protein, consistent with the topology predicted by the cryo-EM model.

## MBOAT7 model reveals how substrates access the enzyme active site

We were unable to obtain a high-resolution cryo-EM structure of MBOAT7 bound to either substrate, possibly due to increased sample heterogeneity. However, we were able to utilize the structure of cMBOAT5 bound to arachidonyl-CoA (PDB 7F40) to model MBOAT7 substrate binding[9]. Docking analyses revealed that arachidonyl-CoA fits well into the cytosolic part of the enzyme tunnel (Fig. 2 and Supplementary Fig. 5), positioning the thioester bond of the substrate near the catalytic His356 and Asn321 (Fig. 2a–c). Asn321 forms a hydrogen bond with the thioester carboxyl bond, implying it stabilizes the carbonyl group during the nucleophilic attack of His356

on the thioester bond. The adenine ring of the CoA moiety protrudes into the cytosol without strong interactions with the protein, similar to the acyl-CoA binding mode of other MBOAT enzymes[10,13,18]. The ribose group of CoA forms hydrogen bonds with Tyr333 (Fig. 2c), and the phosphate groups of CoA form salt bridges with Arg318. The pantetheine group contacts Gln325, Tyr341, Met349 and Ser352, and mutation of Gln325, Tyr341, Ser352 to alanine reduced the acylation activity of MBOAT7 (Fig. 2d). Arg339 faces the entrance of the cytosolic tunnel but does not directly interact with the modeled arachidonyl-CoA. Indeed, mutation of this arginine to alanine did not affect the activity of MBOAT7 (Fig. 2d).

MBOAT7 strongly prefers arachidonyl-CoA as a substrate over other unsaturated fatty acyl CoAs (Supplementary Fig. 1e). The structure provides insight into this preference. The tunnel through MBOAT7 connects to a large cavity next to the catalytic residues, lined with bulky hydrophobic residues, including Phe241, Phe453 and Trp320 (the fatty acid–binding pocket in Fig. 2b, c, colored in green). Trp320 appears to form a hydrophobic gate to exclude water from the catalytic center, and mutating Trp320 to alanine strongly reduced MBOAT7 activity (Fig. 2d). The long, kinked acyl chain of arachidonyl-CoA fits well into this side cavity, interacting with Val313, Tyr407, Met410, Lys403 and Ile369 (Fig. 2c, residues colored in green). In

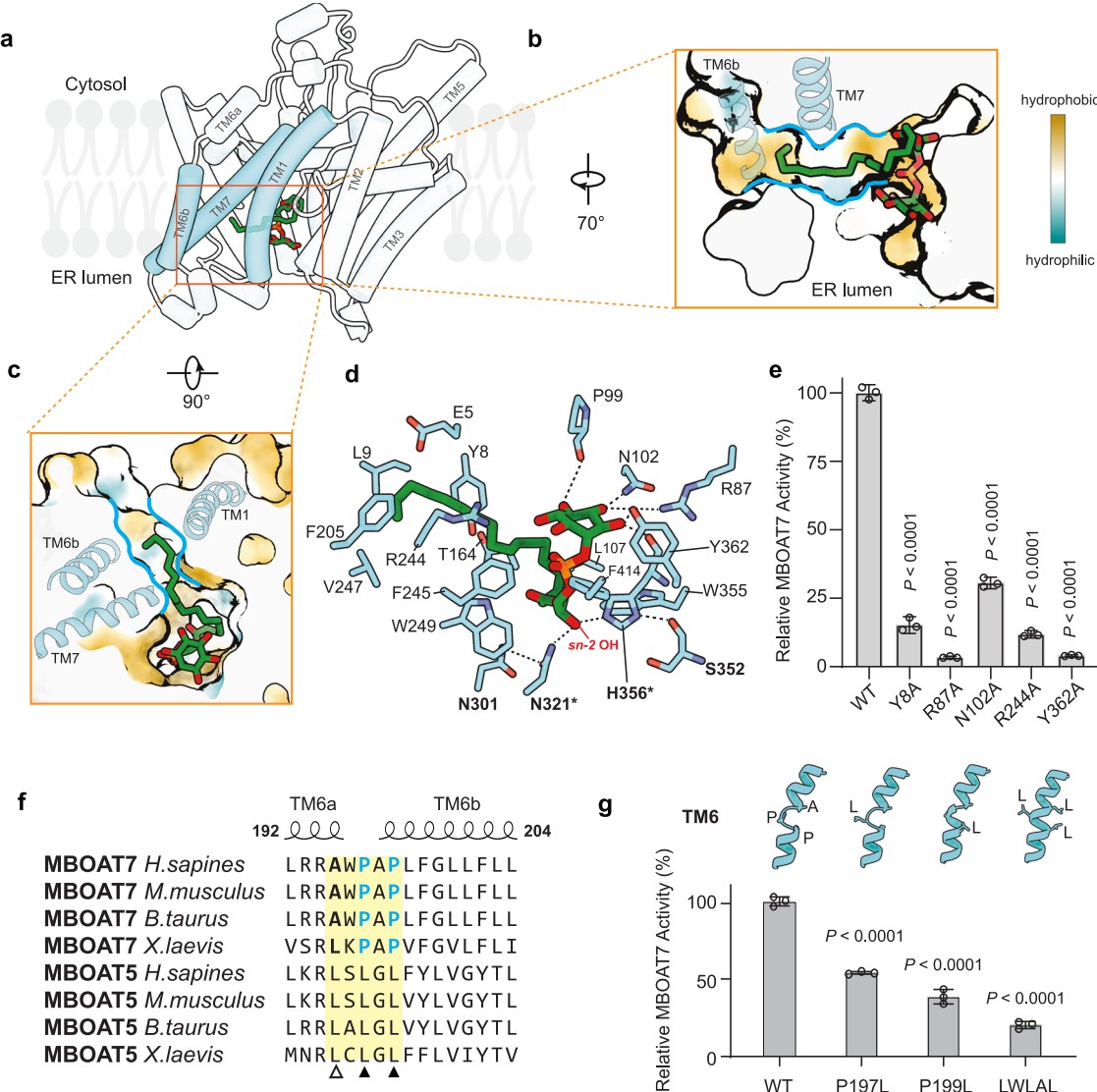

**Fig. 3 | Lyso-PI accesses the MBOAT7 catalytic center through an ER lumenal opening. a** The lumenal access to the enzyme tunnel features an opening for the phospholipid headgroup and a hydrophobic channel for the acyl chain for lyso-PI binding. Cartoon representation of MBOAT7 with 18:0 lyso-PI at the energy minimal state of a molecular dynamic simulation. The helices making up the lateral channel are highlighted in cyan. **b**, **c** Cutting-in views of the lyso-PI molecule in the catalytic chamber, viewed from the ER lumenal side (**b**) or the membrane (**c**). MBOAT7 is represented with hydrophobicity surface. The TM helices comparing the lateral gate were shown in ribbons. **d** Interaction of lyso-PI with MBOAT7 residues. The hydrogen bonds are shown with dashed lines. Residues involved in catalysis are shown in bold, and the two catalytic residues are indicated with asterisks. **e** Activities of the inositol-binding residue alanine mutation. Enzymatic activities of mutants were normalized as the percentage of that of the wild-type MBOAT7

(mean ± SD, $n = 3$ independent experiments). Analysis was performed using one-way ANOVA with Dunnett's *post hoc* test. **f** Sequence alignment between MBOAT5 and MBOAT7 from different species. The secondary structure and residue numbers are shown from the human MBOAT7 apo structure model. The open triangle denotes the evolutionarily variable Ala195, while the solid triangles denote the conserved prolines unique to MBOAT7. **g** Activity of the mutations of the residues at the broken point of TM6. The enzymatic activities of mutants were normalized as the percentage of that of the wild-type MBOAT7 (mean ± SD, $n = 3$ independent experiments). Analysis was performed using one-way ANOVA with Dunnett's *post hoc* test. AlphaFold predictions on structures of MBOAT7 with the corresponding mutations. Ribbon representations of the TM6 are shown only. Source data are provided as a Source Data file.

contrast, straight and saturated acyl-CoAs do not fit this bent cavity well, likely explaining the substrate preference of MBOAT7 for more flexible unsaturated acyl-CoAs.

MBOAT7's tunnel provides a route for lyso-PI to enter the catalytic chamber from the lumenal side of the ER membrane. To gain insight into how specifically lyso-PI binds MBOAT7, we modeled lyso-PI (C18:0) binding to MBOAT7 based on analogy to lyso-PC binding to cMBOAT5 (PDB 7F3X), and analyzed this with all atom-MD simulations. In the resultant energy-minimized structure (Fig. 3 and Supplementary Fig. 6), lyso-PI bound MBOAT7 within the ER lumenal opening of the tunnel. The inositol headgroup occupied the solvent-accessible part of

the tunnel. The *sn-1* acyl chain extended into a hydrophobic side-channel formed by TM1, TM6b, and TM7 (Fig. 3a colored in cyan) that opens into the membrane. This channel is lined primarily with hydrophobic residues (e.g., Leu9, Phe205, Val247, Phe245) that align along the acyl chain of lyso-PI (Fig. 3d). In the apo-MBOAT7 structure, the width of the hydrophobic channel is narrow (~7 Å in diameter) (Fig. 3b, c, hydrophobic channel is highlighted by cyan lines), suggesting that it cannot fit more than one acyl chain. This may explain how MBOAT7 preferentially binds lyso-PI with one acyl chain, but not the vasty more abundant PI with two acyl chains, which could result in product inhibition of the enzyme.

To examine the stability of lyso-PI binding to MBOAT7 and to discover alternative binding modes, we repeated 500-ns MD simulations (Supplementary Fig. 7). Lyso-PI did not leave the binding pocket in each of three simulations. The average RMSDs of MBOAT7 and lyso-PI in the simulations were ~3.5 Å and 7 Å, respectively (Supplementary Fig. 7a), indicating that lyso-PI is stably bound in the pocket. Our simulations also revealed two lyso-PI binding states that appear transiently stable with minor differences to the initial state (Supplementary Fig. 7b, c).

In our structures, the MBOAT7 catalytic residues His356 and Asn321 are close to the lyso-PI glycerol backbone (Fig. 3d). His356 can form hydrogen bonds with the hydroxyl group at the *sn-2* position of the lyso-PI glycerol backbone and the backbone carbonyl group of Ser352 (Fig. 3d). The His356-Ser352 interaction likely increases the electronegativity of the imidazole Nε atom to facilitate the deprotonation of the *sn-2* hydroxyl group of lyso-PI, as found in other MBOAT members[11,12]. Asn321 may also form a hydrogen bond with the *sn-2* hydroxyl group of lyso-PI. Since it also hydrogen bonds with the arachidonyl-CoA thioester carbonyl group (Fig. 2c), Asn321 may assist in the acyltransferase reaction by stabilizing the two substrates close to catalytic His356.

### A discontinuous TM6 helix is important for MBOAT7 activity

Compared with other MBOAT structures, our cryo-EM structure of MBOAT7 is distinct, inasmuch as TM6 is divided into two short helices (Fig. 1d and Supplementary Fig. 4e) linked by two evolutionarily conserved proline residues that are not found in MBOATs 1, 2, or 5 (Fig. 3f and Supplementary Fig. 8). Mutation of the MBOAT7 prolines resulted in a continuous helix in AlphaFold2 predictions and markedly reduced lyso-PI acylation activity (Fig. 3g), suggesting that the discontinuous TM6 is required for optimal MBOAT7 activity.

We obtained a cryo-EM density map of MBOAT7 with lyso-PI and 18:1-ether CoA, a non-hydrolysable acyl-CoA analog (Supplementary Fig. 9). This map provided adequate resolution (~6 Å) to determine the overall architecture and TM arrangement of MBOAT7 with substrates but not the molecular detail. Docking this map with the apo-structure of MBOAT7 identified conformational changes in TM6: in the map obtained in the presence of substrates, TM6b moved away from the lateral portal, and TM6a moved down and became more horizontal. As a result, the lateral hydrophobic channel enlarged, as compared with the apo-structure (Supplementary Fig. 10). We hypothesize that this conformational change facilitates the release of the larger PI product with two acyl chains from the lateral portal, providing a possible explanation for how TM6 mutations reduced the enzyme activity.

### The region between TM4 and TM5 confers lysophospholipid substrate-specificity on MBOATs

The energy minimized structure from MD simulation suggests a unique recognition mode for the inositol headgroup. The inositol ring forms hydrogen bonds with two charged residues, Arg87 and Asn102, and the backbone carbonyl groups of Pro99 and His356 (Fig. 3d). Two charged residues, Arg244 and Tyr8, are close to the phosphate and inositol groups and can transiently interacted with these groups during MD simulations (Supplementary Fig. 7c). Mutation of these residues decreased MBOAT7 catalytic activity (Fig. 3e). Besides Arg244, the phosphate group of lyso-PI is surrounded by a group of aromatic residues, including Tyr362, Phe414 and Phe245 and is stabilized by a combination of hydrogen bond and anion-pi interactions.

How MBOATs 1, 2, 5, and 7 specifically acylate lyso-PS, lyso-PE, lyso-PC, or lyso-PI is unclear. To investigate this, we docked three phospholipid headgroups as ligands into their corresponding MBOAT enzyme structures (obtained from this study or from AlphaFold predictions[32,33]). Each headgroup fits best into a pocket created by the converging ends of two highly variable TM bundles, containing parts

of TM4, TM5, and the connecting loop (Fig. 4a and b), and a more conserved region spanning partial TM9 and TM10 (Fig. 4b). Important residues for recognizing the different headgroups are unique between different MBOATs and almost all clustered on the TM4 and TM5 variable regions (Fig. 4a–c). Specifically, the amino and carboxyl groups in the serine, representing the headgroup of lyso-PS, form salt bridges with three unique charged residues within the variable region in MBOAT1, including His122, Arg125 and Asp137. Choline, representing the headgroup of lyso-PC, is bound by Tyr129 and Cys146 in MBOAT5, a pattern that is similar to the interaction between cMBOAT5 and lyso-PC[9]. For MBOAT7, docking recapitulated Arg87 and Asn102 hydrogen bonding with the inositol headgroup (Fig. 3). Other important residues for headgroup recognition in TM9 and 10 are highly conserved and almost identical in the three MBOAT proteins analyzed (Fig. 4b, c).

To test whether variable regions of TM4 and TM5 determine substrate specificity, we swapped these regions between MBOATs 1, 5 and 7, and tested the activities of the chimeric MBOAT proteins (Fig. 4d, e). Upon replacing the variable regions of MBOAT1 and MBOAT5 with that of MBOAT7, these mutant enzymes gained acyltransferase activity towards lyso-PI (Fig. 4d). Similarly, replacing this region of MBOAT1 or MBOAT7 with that of MBOAT5 yielded enzymes that gained acyltransferase activity towards lyso-PC (Fig. 4d). Thus, these variable regions of MBOATs (corresponding to TM4 and TM5 of MBOAT7) constitute a "specificity-determining region" for different lyso-phospholipids.

### A structure-based screen identified MBOAT7 inhibitors

Increased *MBOAT7* expression correlates with detrimental outcomes in some cancers, including hepatocellular carcinoma and renal clear cell carcinomas, suggesting a dependency on the enzyme[29,30]. Moreover, genetic deletion of *MBOAT7* in clear cell renal carcinoma-derived cells induces cell-cycle arrest and prevents the cells from forming tumors in vivo[29], suggesting MBOAT7-specific inhibitors may be valuable. To identify such inhibitors, we used VirtualFlow[35], a structure-based virtual screening platform, to screen ~5.7 million commercially available compounds (Fig. 5a). We selected candidate compounds, based on differences in free energy–docking scores, molecular scaffolds, and different substructures (Supplementary Fig. 11). Our selection criteria yielded 12 candidates that covered a range of chemical space (Supplementary Fig. 11). We then tested these compounds for inhibition of purified MBOAT7 by measuring activity at two concentrations of compounds, 50 and 5 μM (Fig. 5b). Two compounds, which we term Sevenin-1 and Sevenin-2, inhibited MBOAT7 activity. The IC50s for Sevenin-1 and Sevenin-2 were 0.87 μM [0.56–1.22 μM, 95% confidence interval (CI)] and 5.60 μM (4.61–6.82 μM, 95% CI), respectively (Fig. 5c). Notably, ATR-101[36], a selective ACAT1 inhibitor with multiple reported off-target side effects[37], also inhibited MBOAT7 (Fig. 5b).

To screen for the specificity of Sevenin-1 and Sevenin-2, we measured their effects on purified human DGAT1[13]. Two DGAT1 inhibitors, T863 and DGAT1-IN-1, inhibited DGAT1 activity at 10 μM, whereas Sevenin-1 and −2 showed no inhibition even at 50 μM, suggesting that these two compounds may be MBOAT7-specific (Fig. 5d). According to docking results, both Sevenin-1 and −2 bind to MBOAT7 via occupying the acyl-CoA binding tunnel with the extended side pocket (Fig. 5e, f).

## Discussion

Here, we provide models for the structure of monomeric MBOAT7 and its catalytic mechanism to synthesize PI specifically from arachidonyl-CoA and lyso-PI substrates, and we identify chemical inhibitors of this reaction (Fig. 6). Because it was not possible to obtain a high-resolution structure of substate-bound MBOAT7, we elucidated how MBOAT7 recognizes its two substrates through a combination of molecular modeling, simulations, and biochemical experiments. In this model, the arachidonyl-CoA substrate enters the

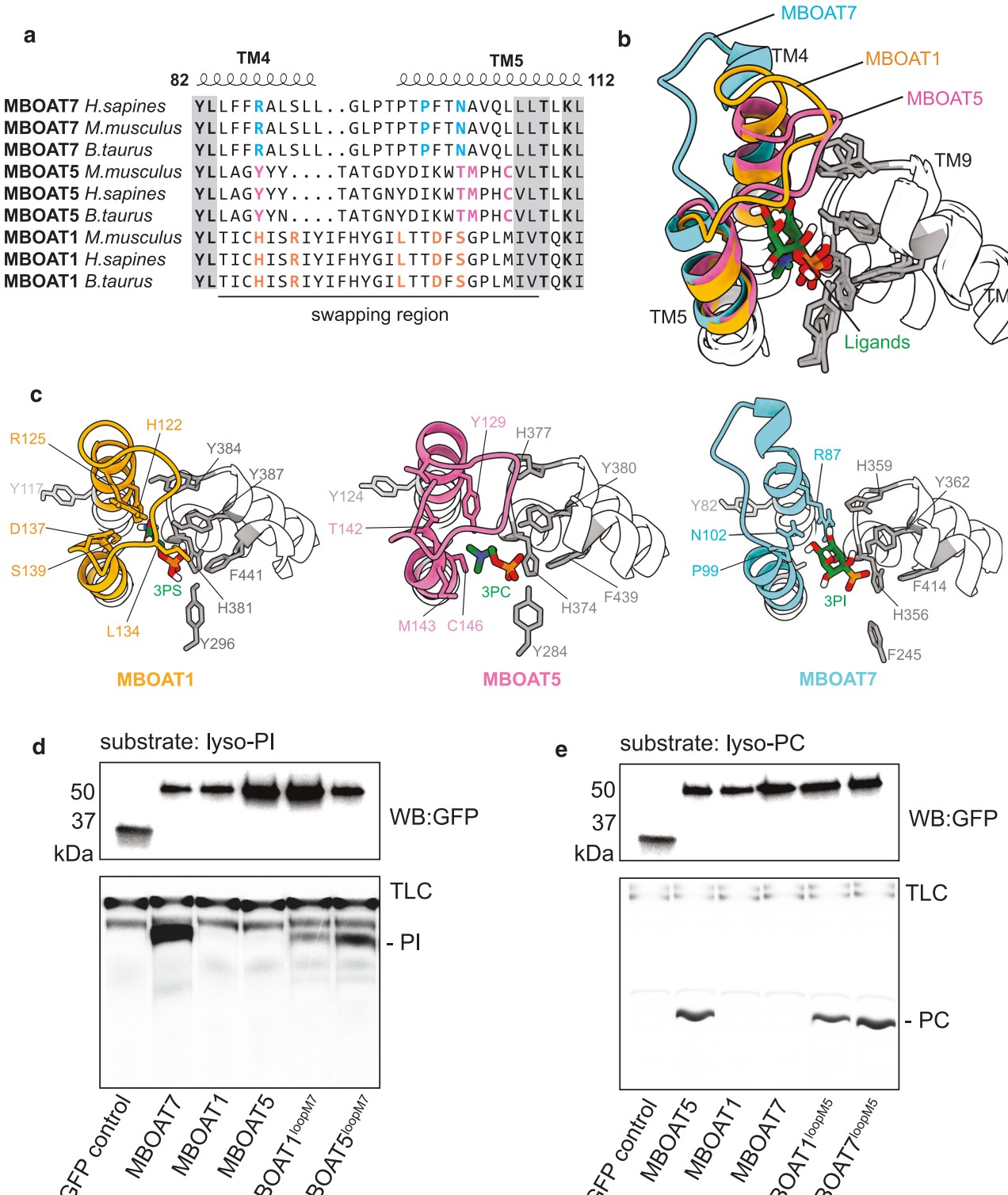

**Fig. 4 | A variable region of MBOAT7 TM segments 4 and 5 determines the selectivity towards lysophospholipid substrates. a** Sequence alignment of MBOAT1, 5 and 7 from different species. Only the variable regions and the short flank sequences are shown. The full sequence alignment can be seen in Supplementary Fig. 8. Conserved residues are highlighted with gray shadow. The key residues for recognizing the specific headgroups were colored. The swapping regions are also indicated. **b** The overlay of pockets for the lyso-phospholipid headgroups from MBOAT1, 2 and 5. The highly conserved residues from TM9 and 10 are shown in gray, and the variable regions are shown in different colors. **c** The detailed interaction between MBOAT and their specific phospholipid headgroups. The colors are consistent with **b**. 3PS, 3-phosphorylserine; 3PC, 3-phosphorylcholine; 3PI, 3-phosphorylinositol. **d, e** The activity analysis of the region-swapped chimeric proteins. Protein levels were adjusted to the immunoblotting against GFP (upper). The activity is detected by the production of radioactive PI (**d**) or PC (**e**) on a TLC. The experiments were independently performed 3 times with similar results. Source data are provided as a Source Data file.

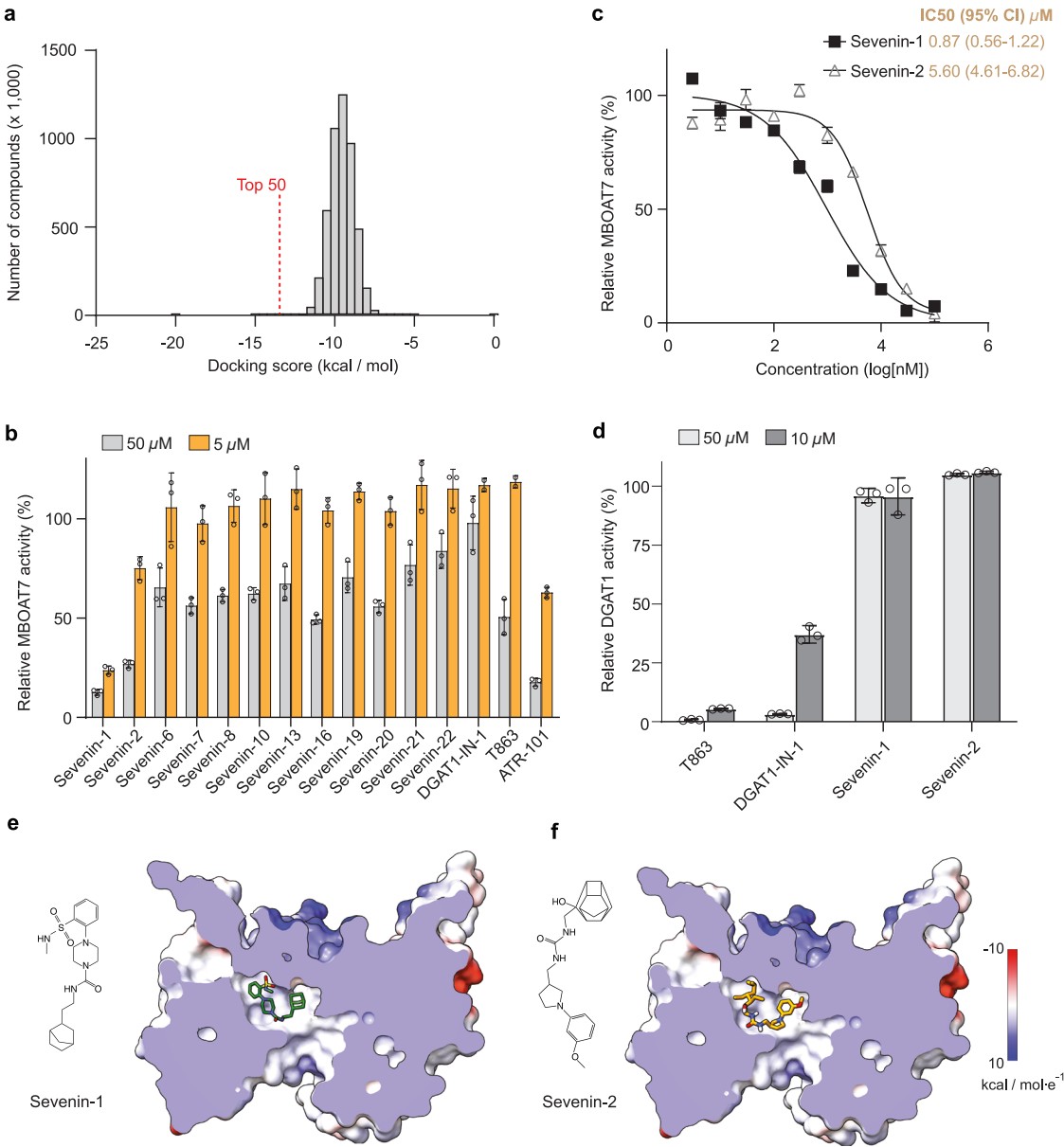

**Fig. 5 | Structure-based discovery of MBOAT7 inhibitors. a** Histogram plot of the screen -5.7 million compounds based on their free-energy docking score. **b** Relative MBOAT7 activity with the addition of indicated compounds at 50 or 5 μM. Sevenin-number; the number indicates the compound ranking from the VirtualFlow screen. Activities were normalized to vehicle control (mean ± SD, $n$ = 3 independent experiments except $n$ = 2 for DGAT1-IN-1 and T863 at 5 μM). **c** IC50 curves for inhibition of MBOAT7 activity by Sevenin-1 and −2 measured by radioactivity-based assays (mean ± SD, $n$ = 3 independent experiments). **d** Inhibition of human DGAT1 activity by different compounds (mean ± SD, $n$ = 3 independent experiments). **e, f** Chemical structures of Sevenin-1 (**e**) and Sevenin-2 (**f**), and their docking positions in the MBOAT7 structure. MBOAT7 is shown with an electrostatic surface. Source data are provided as a Source Data file.

enzyme tunnel from cytoplasmic leaflet of the ER. An extended lateral side pocket can accommodate the flexible acyl chain of unsaturated acyl-CoAs (Fig. 6a), but is not optimal for a straight, saturated acyl chains. Lyso-PI, enters the tunnel from the lumenal leaflet with a side channel harboring its single acyl chain and positions the *sn*−2 hydroxyl group of the glycerol backbone near the catalytic His356 residue. In the putative catalytic mechanism, His356 deprotonates the hydroxyl group of the glycerol backbone, enabling the oxygen to undergo a nucleophilic attack on the carbonyl group of the thioester bond (Fig. 6b). After forming an intermediate, electron transfers would result in formation of the CoA-SH and PI products. We hypothesize that binding of the substrates causes conformational changes in the TM6 helix to widen the hydrophobic, lumenal opening of the enzyme, allowing release of PI, with two acyl chains, into the membrane.

A key question for MBOAT enzymology is how substrate specificity is achieved. Particularly, the phospholipid remodeling MBOAT enzymes have a high degree of evolutionary sequence conservation, are very similar in structure, and yet are able to distinguish between different lyso-lipid substrates. Our data suggest that TM segments 4–5 of MBOAT7 (or equivalent sequences of other Lands cycle MBOATs) constitute a lyso-phospholipid "specificity-determining region". Key residues in this region are conserved for each different MBOAT and appear to interact with each headgroup (Fig. 4). Indeed, our analysis of chimeric expressed proteins indicate that these regions determine specificity for PI versus PC for MBOATs 5 and 7.

Beyond Lands cycle enzymes, phylogenetic analyses of MBOATs clusters them into groups according to their substrates (e.g., hydrophobic neutral lipids, hydrophilic polypeptides, or amphipathic lyso-

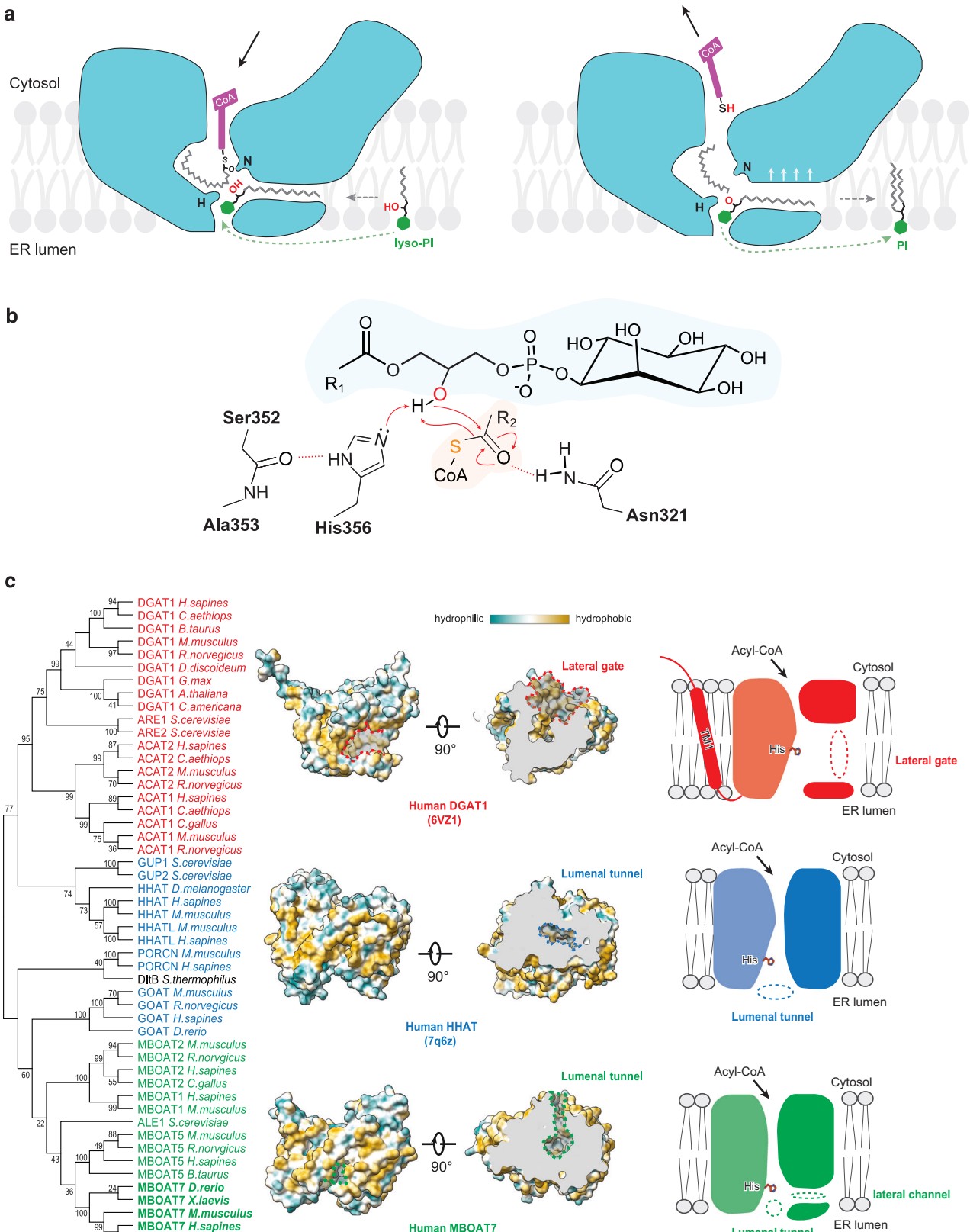

phospholipids) (Fig. 6c). Whereas the acyl-CoA binding regions of MBOATs are relatively conserved, different paths appear to allow acyl-acceptor substrates to access the catalytic site (Fig. 6c). For MBOATs that acylate proteins (e.g., hedgehog acyltransferase, porcupine), the protein substrate reaches the catalytic center from the ER lumen. For MBOATs that catalyze neutral lipid synthesis (e.g., DGAT1, ACAT1,

ACAT2), the substrate accesses the catalytic center from a lateral gate in the plane of the membrane. For the MBOATs that mediate phospholipid remodeling, such as MBOAT7, the lysophospholipid substrates access the enzyme tunnel from the lumenal side.

Loss-of-function mutations in human *MBOAT7* result in liver disease, intellectual disability, early onset seizures, and autism spectrum

**Fig. 6 | A model for MBOAT7 catalytic function and comparison of general MBOAT classes. a** Hypothetical model for MBOAT7-catalyzed PI remodeling. The membrane-partitioned or soluble acyl-donor arachidonyl-CoA enters the reaction center at the MBOAT7's cytosolic tunnel. The side pocket accommodates the long and kinked arachidonyl acyl chain. The acyl-acceptor lyso-PI enters the catalytic chamber through the connected lateral channel and ER lumenal tunnel, with the hydrophobic acyl chains enters through the lateral channel (gray dashed arrow), and the hydrophilic inositol headgroup slides in through the ER lumenal tunnel (green dashed arrow). The thioester bond and *sn-2* hydroxy group of lyso-PI are positioned close to the catalytic Asn321 (N) and His356 (H). Once the catalysis is complete, CoA-SH is released into the cytosol, and PI is released to the ER lumenal leaflet of the membrane. The lateral gate is opened slightly wider to facilitate the rapid release of PI. **b** Putative chemical mechanism of MBOAT7-catalyzed PI remodeling. The backbone carbonyl of Ser352 forms hydrogen bonds with the N on the imidazole ring, which polarizes the other N to deprotonate the sn-2 hydroxyl group of lyso-PI glycerol. Deprotonated hydroxyl initiates a nucleophilic attack on the thioester bond of the arachidonyl-CoA. Finally, CoA-S$^-$ attracts the proton back from His356 to complete the catalytic cycle. **c** Phylogenetic analysis and structural comparison of MBOATs in different clusters. MBOATs with different substrate preferences were clustered and highlighted with different colors except the bacteria DltB. Red, blue and green indicate preference towards neutral lipids, polypeptide, and lyso-phospholipids, respectively. One representative structure from each cluster is shown with hydrophobicity surfaces. The key features for acyl acceptors to access are highlighted in dashed lines and labels. Cartoons are also shown on the right. Dashed circles denote the different entry sites of the acyl-acceptor substrates.

disorders[19–21,25–28,30]. Mapping the pathogenic *MBOAT7* mutations responsible for these disorders on to our MBOAT7 structure model enables some predictions on how they disrupt MBOAT7 function (Supplementary Fig. 12a). For example, the E253K disease-causing mutation[19], though located in TM7 away from the catalytic center, may destabilize the protein by disrupting hydrogen bonds of Glu253 with the backbones of Asn301 and Ile 302 (Supplementary Fig. 12b). The Arg384 mutation, found in patients with intellectual developmental disorders, is close to the entrance of acyl-CoA binding pocket and predicted to interfere with the recognition or binding of acyl-CoA substrates (Supplementary Fig. 12c).

High expression of *MBOAT7* has been detected in multiple types of cancer[29,30], and studies of clear cell renal carcinoma suggest MBOAT7 inhibition as a possible therapy[29]. We therefore thought to utilize the MBOAT7 structure to identify lead inhibitory compounds. Indeed, a structure-aided virtual screen identified two potential inhibitors, Sevenin-1 and −2 (Fig. 5). Although these compounds require further characterization and optimization, they provide a foundation for developing selective MBOAT7 inhibitors as therapeutic agents for aggressive cancers with MBOAT7 dependency, where their use may be valuable despite potential adverse effects such as causing transient fatty liver. Interestingly, similar to other MBOAT inhibitors[18,38,39], the structures of Sevenin-1 and −2 contain two bulky aromatic groups connected by an amide bond. This could be a common scaffold for the MBOAT inhibitors, which may help guide the design of future inhibitors for these enzymes.

## Methods
### Protein expression and purification
MBOAT7 from *Homo sapiens* (UniProt ID: Q96N66) was expressed in HEK293 freestyle cells (ThermoFisher), using the inducible stable cell line system[40]. Briefly, the MBOAT7 cDNA sequence was cloned into the pSBtet vector with a C-terminal GFP-strepII sequence. The stable cell line was generated by co-transfection of the pSBtet-MBOAT7 and the SB-100X transpose vector, followed by a week of puromycin selection. The stable cells were grown in suspension at 37 °C in FreeStyle 293 Expression Medium (ThermoFisher). When the cells reached a density of $2.0–2.5 \times 10^6$ cells per ml, 20 μM doxycycline and 5 mM sodium butyrate were added to the cells to induce the protein production and boost the expression, respectively. Temperature was set to 30 °C. The cultures were harvested after ~60 h. The cells were washed once in ice-cold PBS, then flash frozen in liquid nitrogen and stored in −80 °C or placed on ice for immediate use.

All protein purification steps were performed at 4 °C. Thawed cell pellets were re-suspended in the lysis buffer containing 50 mM Tris-HCL, pH 8.0, 150 mM NaCl, 1 mM EDTA and supplemented with 1× complete protease inhibitor cocktail (Roche). DDM/CHS detergent (1%/0.1%) (Anatrace) was added to lyse the cells for 2 h. Insoluble debris was removed by centrifugation $50,000 \times g$ for 45 min, and the supernatant was incubated with prewashed Strep-Tactin Sepharose beads (IBA) for 1 h. The resin was then washed with ~20 ml of washing buffer (lysis buffer containing 0.1% digitonin) before the elution of GFP-fused MBOAT7 protein by washing buffer containing 2 g/L des-thiobiotin. TEV proteases were then added to the eluted proteins to remove the GFP and strep tag before the elution was concentrated by a 50-kDa MWCO spin concentrator and injected onto a Superose 6 column (GE Healthcare) equilibrated in 100 mM Tris-HCl, pH 7.5, 150 mM NaCl, and 0.05% (w/v) digitonin. The peak fractions containing MBOAT7 were pooled and concentrated. To reconstitute MBOAT7 proteins in PMAL-C8 (Anatrace), MBOAT7 in detergent was mixed with PMAL-C8 at a 1:3 (w:w) ratio, followed by gentle agitation overnight in the cold room. Detergent was then removed by mixing with Bio-Beads SM-2 (Bio-Rad) for 1 h. Bio-beads were removed by passing through a disposable polyprep column. The MBOAT7 protein was further purified by injecting onto a Superdex 200 increase 10/300 GL column equilibrated with 100 mM Tris-HCl, pH 7.5, and 150 mM NaCl.

To prepare the protein sample with lyso-PI and acyl-CoA, 10 μM lyso-PI was added at each step after cell solubilization and before PMAL-C8 reconstitution, whereas 10 μM acyl-CoA or nonhydrolyzable analogs was added at each step before the final size-exclusion purification.

### Cryo-EM imaging and data processing
Grids for cryo-EM imaging were prepared on a Vitrobot Mark IV system (ThermoFisher Scientific) by applying 3 μL of PMAL reconstituted MBOAT7 at 5–6 mg/ml to Quantifoil holey gold grids (Au R1.2/1.3; 400 mesh) that were glow discharged for 30 s at 15 mA right before use. Grids were blotted for 5 s at 4 °C, 100 % humidity, and a blot-force of 12 N before being plunge frozen in liquid ethane cooled by liquid nitrogen. Cryo-EM data were collected on a Titan Krios electron microscope with a K3 direct electron detector. Movies were collected in counting mode with a pixel size of 0.825 Å and a total does 50.4 electron/Å2.

Movies were corrected for beam-induced motion by MotionCor2[41], and contrast transfer function parameters were determined using CTFFIND4[42]. Template particle picking and 2D classification were performed using Simplified Application Managing Utilities of EM Labs (SAMUEL) similarly to previous dataset processing[43]. To identify particles with rare angles, the deep-learning-based particle picker Topaz[44] was used and followed by multiple rounds of 2D classifications in cryoSPARC[45]. Selected particles from SAMUEL and cryoSPARC were merged, and the redundant particles were removed before 3D processing with Relion3.0[46], including 3D classification, 3D refinement, and domain masking. 3D classification on binned particles at 3.3 Å pixel size were first performed to remove bad particles, and additional rounds of 3D classification were performed on less binned and unbinned particles. 3D classes were combined or discarded based on the quality of cryo-EM density of the transmembrane helices. Overall resolution of cryo-EM map was computed according to the gold standard Fourier shell correlation (FSC) method. The local resolution was calculated with the local resolution estimation function in

Phenix[47]. The directional resolution of the cryo-EM map was evaluated by the 3D FSC spherical score using the web server (https://3dfsc.salk.edu/)[48]. For the 3D FSC calculation, a protein mask was provided in addition to the final map and two half maps. A flow-chart summary on the data processing can be found in Supplementary Fig. 2. The statistical details related to data processing and map validations are summarized in Supplemental Table 1.

## Model building and refinement

The AlphaFold 2 predicted MBOAT7 model was used as the initial model. The model was docked into the map and refined using phenix.real_space_refine in Phenix[47] and manually adjusted in COOT[49]. The resulting map was further put back through the real-space refinement procedure to undergo further refinement. The final model was validated with MolProbity[50]. The map-model fitness was evaluation based on the Q score[51], which is summarized in Supplemental Table 1. The comparison between the initial Alpha-Fold model and final MBOAT7 apo model is shown in Supplementary Fig. 3.

## Mass spectrometry

To confirm substrate binding with purified MBOAT7 during protein preparation, lipid contents were extracted using the tert-butyl-methyl ether and methanol method[52] from purified MBOAT7 proteins and lipid controls (Avanti Polar Lipids). Extracted lipids were resolubilized in 300 μL of isopropanol, methanol and chloroform (4:2:1, v:v:v) containing 7.5 mM ammonium acetate. Mass spectrometry analysis of lipid mixture was performed using an Orbitrap ID-X Tribrid mass spectrometer (ThermoFisher Scientific), equipped with an automated Triversa Nanomate nanospray interface (Advion Bioscience) for direct infusion lipid solution delivery. All full scan mass spectra were acquired in negative mode using mass resolution of 500,000 (FWHM at m/z 200). Visualization of mass spectrometry data was performed using Qual Browser on Xcalibur 4.3.73.11 software (Thermo Scientific).

## Molecular dynamics simulation and docking

MBOAT7 was placed in a POPC bilayer by using the orientations of proteins in membranes (OPM) database (https://opm.phar.umich.edu/) as a reference and an in-house protocol. Briefly, proteins and lipids were prepared at Martini resolution, followed by short equilibrium. The Martini protein was then replaced with the original MBOAT7 apo structure and the Martini lipids were back-mapped into the membrane. Lyso-PI was modeled based on lyso-PC bound to cMBOAT5 (PDB 7F3X). The topology of lyso-PI was made based on the topology of PI and diacylglycerol and is available at git@github.com:ksy141/TG.git. One round of energy minimization resulted in the energy-minimized structure. Simulations were carried out three times, each of which was 500 ns long. Simulations were performed using the GROMACS (version 2018)[53] simulation engine with the CHARMM36m lipid and protein force field[54,55]. Simulations were integrated with a 2-fs timestep. Lennard-Jones pair interactions were cut off at 12 Å with a force-switching function between 10 and 12 Å. The Particle Mesh Ewald algorithm was used to evaluate long-range electrostatic interactions[56]. The LINCS algorithm was used to constrain every bond involving a hydrogen atom[57]. A temperature of 310 K and a pressure of 1 bar were maintained with the Nose-Hoover thermostat and the Parrinello-Rahman barostat, respectively[58-60]. The coupling time constants of 1 and 5 ps were used, respectively. A compressibility of $4.5 \times 10^{-5}\,\text{bar}^{-1}$ was used for semi-isotropic pressure coupling. MDAnalysis was used for analyzing the trajectories[61].

The constraint docking was performed using the MedusaDock 2.0 program[62]. The headgroup search space were estimated based on cMBOAT5 and MBOAT7 structures. Constraints were set between the phosphate group and conserved interacting residues.

## Fluorescence protease protection assay

Membrane topology of MBOAT7 constructs was determined by fluorescence protease protection assay, as described previously[34]. In brief, MBOAT7 and control (mCherry-ER3 and mEmerald-Sec61β) constructs was transfected into the SUM159 cells using the FuGENE® Transfection Reagent according to the manufacturer's protocol. Two days post transfection, SUM159 cells were washed three times with 2 ml of KHM buffer (110 mM potassium acetate, 20 mM HEPES-KOH pH 7.2, 2 mM MgCl$_2$) at room temperature. During the imaging acquisition, 40 μM digitonin and 100 μg/ml proteinase K (Sigma) were added to the media at the indicated time points for plasma membrane permeabilization and fluorescent protein degradation, respectively. SUM159 breast cancer cells were obtained from the laboratory of Tomas Kirchhausen at Harvard Medical School (ATCC, CRL-1504) and were maintained in DMEM/F-12 GlutaMAX (Life Technologies) with 5 μg/ml insulin (Cell Applications), 1 μg/mL of hydrocortisone (Sigma), 5% FBS (Life Technologies 10082147, Thermo Fisher), 50 μg/mL of streptomycin, and 50 U/mL of penicillin.

## MBOAT7 mutagenesis and activity assays

The Michaelis-Menten and substrate specificity assays were performed using purified and PMAL reconstituted MBOAT7 with a fluorescence-based coupled-enzyme assay, as described[12,14]. Essentially, the activity was monitored by detecting the release of the free CoA-SH with the fluorescence probe 7-diethylamino-3-(4-maleimidophenyl)−4-methylcoumarin (CPM). A 10-μL reaction mixture contains 2 mg/ml BSA, 75 mM Tris, pH 7.5, 100 nM methyl arachidonyl fluorophosphonate (MAFP), 150 mM NaCl, and 0.025 g/L of MBOAT7 protein and the substrates acyl-CoA and lyso-PI with varied concentrations. For Michaelis-Menten characterization, either arachidonyl-CoA or lyso-PI was maintained at 100 μM, and the concentrations other substrate were varied. For characterizing substrate specificity, different acyl-CoA and lyso-phospholipids were added to a final concentration of 100 μM. The reaction was initiated by adding the enzymes to the reaction. The reaction proceeded for 3 min at 37 °C before being stopped by adding 1.3% SDS and incubating at room temperature for 5 min. 90 μL of CPM working solution (50 μM in 75 mM Tris, pH 7.5 and 150 mM NaCl) was added to the reaction, and the mixture was kept at room temperature for 30 min and followed by fluorescence detection (excitation 355 nm; emission 460) using the Tecan i-control infinite 200 plate reader. The background was determined from reactions without adding enzymes and subtracted to remove the background noise. MBOAT7 activity was stable over 15 min in the reaction buffer. Nonlinear regression to the Michaelis-Menten equation and allosteric sigmoidal analysis were performed using GraphPad Prism 9.

MBOAT7 mutants were generated by site-directed mutagenesis on the pFBM construct expressing MBOAT7 with a C-terminal GFP and strep tag using the Q5 Mutagenesis Kit (New England Biolabs), according to the manufacturer's protocol. The primers used for mutagenesis are in Supplementary Data 1. The sequenced constructs were transfected into the HEK293 freestyle cells with PEI, and the cells were harvested after 70 h incubation at 30 °C. The cells were lysed and proteins were purified as described above with the modification that DDM/CHS was used for all solubilization, purification, and elution steps, and 10% glycerol was added in the final elution buffer. The initial levels of the wild-type and mutant MBOAT7 proteins were measured by immunoblotting against the GFP tag fused to the C-terminal of MBOAT7 and adjusted to the same quantity for assays. The acyl-transferase activity was determined by measuring the incorporation of radiolabeled oleoyl-CoA into the product phosphatidylinositol[6]. In brief, a total 100 μL reaction mixture was prepared containing 75 mM Tris-HCl, pH 7.4, 2 g/L delipidated BSA, 150 mM NaCl, 100 nM MAFP, 50 μM 18:0 lyso-PI, 100 μM oleoyl-CoA containing 0.2 μCi [$^{14}$C]-oleoyl-CoA as a tracer (American Radiolabeled Chemicals), and proteins (free

GFP control, WT or mutant MBOAT7, -100 ng) to initiate the reaction. The reaction was started by adding proteins to the reaction mixture and incubated for 15 min at 37 °C. The reactions were quenched by adding chloroform/methanol (2:1 v:v), followed by 2% phosphoric acid for phase separation. The organic phase was harvested, dried, resuspended in chloroform and loaded on a silica gel TLC plate (Analtech). Lipids were resolved in a solvent system consisting of chloroform, methanol, acetic acid, and water (65:35:8:5 v:v:v:v). The radioactivity of the phosphatidylinositol bands was revealed by Typhoon FLA 7000 phosphor imager (GE Healthcare Life Sciences) and quantified by Quantity One (V4.6.6).

## VirtualFlow drug screening
The structure-based drug screen was performed essentially according to the instructions on the VirtualFlow website (https://virtual-flow.org/) and github page (https://github.com/VirtualFlow/VFVS). Firstly, the searching space was determined by covering both the cytosolic and ER lumenal tunnels of MBOAT7 apo structural model using the AutoDockTools[63]. 5.7 million ligands were selected from the REAL library based on the properties of available small-molecule inhibitors developed for other human MBOAT members (DGAT1, ACAT1/2, Porcupine, HHAT) with the following criteria: molecular masses 375–425 Daltons; partition coefficient (SlogP) 1–2.5, hydrogen bond acceptors (HBA) 3–5; hydrogen bond donor (HBD) 1–2. The dockable forms of the 5.7 million compounds were directly downloaded from the website (https://virtual-flow.org/real-library). The VirturalFlow screen was performed using QuickVina program and each compound was docked once. The process took ~6 weeks using 100 cpu cores. The final 50 compounds with the highest docking scores (the lowest free energy) were analyzed based on their fingerprint similarity using the python RDKit package (https://www.rdkit.org/) and classified by signature substructures (Supplementary Fig. 11). Then, 12 compounds were further demanded from the Enamine company and tested in vitro. For some compounds, the chemical structures of the synthesized compounds and original docked compounds in VirtualFlow are slightly different possibly due to the file conversion during library preparation or library update in Enamine.

## Reporting summary
Further information on research design is available in the Nature Portfolio Reporting Summary linked to this article.

## Data availability
The three-dimensional cryo-EM density maps have been deposited into the Electron Microscopy Data Bank under accession numbers EMD-28552. The coordinates are deposited into the Protein Data Bank with accession number 8ERC. This study also cited published structures 7F3X (chicken MBOAT5 in complex with lyso-PC) and 7F40 (chicken MBOAT5 in complex with arachidonyl-CoA). The raw TLC, gel and blot data generated in this study are provided in the Source Data file. Source data are provided with this paper.

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

## Acknowledgements

We thank S. Sterling and M. Mayer at the Harvard cryo-EM center, and C. Xu and K. Song at the University of Massachusetts cryo-EM facility, for electron microscopy data collections and Z. W. Lai for help with mass spectrometry experiments. The MD simulations were performed on the high-performance GPU cluster (GM4) at the University of Chicago Research Computing Center (supported by NSF grant DMR-182869). The VirtualFlow drug screen was performed using the high-performance CPU clusters at the Harvard Medical School O2 system. We are grateful for discussions with the members of the Liao and Farese & Walther laboratories, and we thank Gary Howard for editorial assistance. S.K. and G.A.V. were supported by R01GM063796. This work was supported by R01GM141050 (to R.V.F.) and the Howard Hughes Medical Institute (T.C.W).

## Author contributions

K.W., M.L., T.C.W. and R.V.F. conceived the project. K.W. performed protein expression, purification, cryo-EM grid preparation, data processing, model building, molecular docking and VirtualFlow drug screen studies. M.L., S.W. and X.S. advised on cryo-EM data processing. K.W., C.-W.L., and A.B.H. performed the mutagenesis and activity studies. C.W.L. performed the fluorescence protease protection assay. S.K. and G.A.V. performed molecular dynamics simulations. A.J.B. performed the mass spectrometry analysis. K.W., M.L., T.C.W. and R.V.F. wrote the manuscript. All authors analyzed and discussed the results and contributed to the manuscript.

## Competing interests

The authors declare no competing interests.
