## [Peer Review File · Nature Communications]

The structure of phosphatidylinositol remodeling MBOAT7 reveals its catalytic mechanism and enables inhibitor identificationREVIEWER COMMENTS

Reviewer #1 (Remarks to the Author):

MBOAT 7 enzyme plays important role in lipid metabolism. It also plays important role in controlling membrane lipid composition. The current manuscript provides a careful and detailed analysis of the biochemical and structural analysis of MBOAT7. The results are compelling. I have a few questions. They need to be carefully addressed in the revised manuscript.

1. The site for binding the substrate LPI was not elucidated. Please carefully address this shortcoming. Please indicate what needs to be done in order to solve the problem.
2. The authors look for a small molecule inhibitor of MBOAT7. MBOAT7 plays a key role in remodeling the FA composition of multiple phospholipids. The rationale for finding small molecule inhibitors is not clear to this reviewer. Please indicate why inhibiting MBOAT7 is a good thing. Why not look for activators of MBOAT7? Is the MBOAT7 gene known to be regulated by SREBP or LXR? Please briefly discuss.
3. The cryo-EM of MBOAT7 has been published last year. Please indicate what new information the current manuscript offers.
4. The authors describe the biochemical enzyme assay for MBOAT7. The assay time is 7 minutes. What is the rationale? Please provide the ref for this assay. Is the enzyme stable in the assay condition described?
5. ATR-101 is a potent ACAT1 inhibitor. It is clinically approved to treat a rare form of cancer in the adrenals. Here the authors write ATR101 as a prototypical ACAT1 inhibitor. Please note that ATR-101 produces multiple off-target side effects as reported by Langlois et al published at PMID: PMC5932779. DOI: 10.1186/s12902-018-0251-5. Other ACAT1 inhibitor does not possess the same phenotypes as ATR101.

Reviewer #2 (Remarks to the Author):

The focus of this manuscript is the membrane-bound O-acyltransferase 7, MBOAT7, responsible for acetylating lyso-PI with arachidonyl-CoA. In this work, Wang et al. present the structure of MBOAT7 determined by cryo-EM, alongside new mechanistic insights and determinants of phospholipid headgroup selectivity. MBOAT7 is a therapeutic target and an in silico screen presented here resulted in the identification of small-molecule inhibitors of MBOAT7.

Key findings of this work are that

- i) The TM6 of MBOAT7 was found to be a discontinuous helix, broken by Pro residues, the mutation of which led to reduced lyso-PI acylation activity. A lower resolution substrate-bound MBOAT7 cryoEM structure was used to demonstrate a shift in TM6, providing more support for a mechanistic role of the discontinuous TM6 helix.
- ii) Loop-swap mutations of the TM4-TM5 region allowed the phospholipid selectivity of MBOATs to be altered, providing new insight into the recognition of lipid headgroups
- iii) Molecular simulations were used to characterise MBOAT7-lyso-PI interactions, where the sn-1 acyl chain was observed to be extended into a hydrophobic channel
- iv) A structure-based screen led to two compounds that were found to inhibit MBOAT7 activity

Overall, this work represents a significant advance in our understanding of MBOATs involved in the Lands cycle and is valuable for our understanding of lipid specificity across other systems.

This reviewer has two main concerns to be addressed:

- a. The model used for the basis of molecular simulations has not been well-described. How was lyso-

PI modelled into the cavity? It is not possible to judge the simulation quality as the only data presented from the simulation is an 'energy-minimised structure', it is not clear what this means in this context. Is this a representative conformation from the independent 500 ns simulations? It would be more valuable for the reader if more information was included, such as how the initial model was built and snapshots of the initial and final orientations of lyso-PI. It would also be of interest to see if the TM6 conformation changed in the presence or absence of lyso-PI during simulations. It was not described in the Methods how MBOAT7 was modelled into the POPC membrane.

b. There is very little information on the VirtualFlow screening. More detail is needed on how the 12 candidate compounds were selected. How were scaffold similarity criteria incorporated into this decision?

This manuscript is very well presented and there are few minor edits:

c. Extended Data Fig 4 caption, "corresponding residues are labelled in pair" unclear meaning, could be rephrased for clarity

d. Extended Data Fig 5 what lyso-PI structure? Is this a representative model from the molecular dynamics simulations presented here?

e. Discussion of the possible basis of disease-causing mutations E353K and R384Q based on this new structure – this could be clearer for the reader if panels showing this were incorporated into Extended Data Fig. 9.

Reviewer #3 (Remarks to the Author):

The manuscript under consideration presents the structure of the membrane-bound O-acyltransferase 7 (MBOAT7), which catalyzes the acylation of lyso-phosphatidylinositol (lyso-PI) with arachidonyl-CoA. The structure was obtained with single-particle cryoEM of amphipol-embedded (PMAL-C8) human MBOAT7 produced in HEK293 cells, leading to a reconstruction of 3.7Å global resolution. The authors use a fluorescence-based assay and a TLC assay to characterize the substrate specificity of the enzyme towards different donors and acceptors for the catalyzed acyl chain transfer reaction.

The authors were unable to obtain high-resolution structures of substrate bound MBOAT7, thus they proceeded to approximate the substrate-bound states using computational methods. In the case of arachidonyl-CoA (donor) they superimpose the MBOAT7 structure with that of cMBOAT5 bound to arachidonyl-CoA. For, Lyso-PI, the authors obtain a low resolution reconstruction to identify conformational changes within the transmembrane domain of MBOAT7, and additionally, they perform molecular dynamics simulations to characterize how the acceptor substrate fits within the acceptor cavity. Importantly, in both cases the authors pair their computational predictions with mutagenesis experiments that appear to confirm the substrate binding predictions.

The workflows used for cryoEM structure determination are presented clearly, and have sufficient detail. The statistics presented for the structure are appropriate for the resolution ranges being reported. Importantly, the densities presented on extended data fig. 2 are consistent with the reported resolution. The presentation of a local resolution map and an angular distribution plot for the dataset was appreciated.

Overall, the manuscript represents high-quality and medium to high-impact work that advances our understanding of phospholipid biogenesis and modification. The manuscript is well-written and the methods section is sufficiently detailed to allow replication.

Some points for improvement are listed below:

1) Line 149: It is mentioned that mutation of Met349 to Ala leads to reduced acylation activity, yet this residue is not present in the bar graph shown in Fig. 2d

2) In Extended fig 8 the two states of the protein (apo and lyso-PI – bound) are shown at low resolution for comparison. In text (line 196), it is mentioned that the lateral hydrophobic window is enlarged in the lyso-PI-bound state. Adding an alternative view like in fig 3c, for apo vs bound, would highlight better the purported difference.

3) The authors have used the Alphafold2 predicted model as an initial model for constructing their model based on the cryoEM reconstruction. I believe that a comparison between the starting (Alphafold2) model and the final (cryoEM) model would be of great interest to readers as a test case.

4) On Fig. 2c a visual grouping (something like a colored bracket) of the residues that contribute to the fatty acid binding pocket could help the reader to better orient themselves within the presented active site (correspondence between Fig. 2b and 2c)

5) For the model validation a MolProbity score and Clashscore is provided. It would be nice to also see some map to model type validation like the Q score that is included in the PDB validation report or an EM ringer score. A description of the score(s) used should be included in the methods section. Finally, a 3DFSC sphericity score should be included for quality assessment of the data.

Reviewer #4 (Remarks to the Author):

The paper “The structure, catalytic mechanism, and inhibitor identification of phosphatidylinositol remodelling MBOAT7” describes novel structure of MBOAT7. The protein functionality has been confirmed by biochemical and biophysical methods, mutagenesis and chimeric-proteins modifications. Very detailed analysis of the structure and substrate docking allowed to suggest mechanics of possible substrate binding and its specificity. Authors speculate on the effect of disease related mutations based on their position on the structure. Paper is well written and easy to follow, conclusions are justified, experimental methods have enough details to reproduce the methodology. The paper contains high quality illustrations and detailed methodology. I recommend publication after minor corrections.

Comments:

It is known that loss-of-function mutations in human MBOAT7 results in liver disease, intellectual disability, early onset seizures, and autism spectrum disorders” Could use in Inhibitors like Sevenin-1 and -2. Use of Inhibitors will be equivalent to the “loss of function”. Will protein inhibition cause similar diseases as mutations with the loss of function? Silencing the gene also seems lethal. I would like authors to add extra thoughts on benefits of inhibitors and how to avoid mutation-like effects by using inhibitors.

Minor comments

1. please check if you refer to correct figure

Line 122 “multiple conformations in the apo state of the enzyme (Extended Data Fig. 2f).”
Figure 2f shows density with fitted helices, please refer to correct figure.

REVIEWER COMMENTS

Reviewer #1 (Remarks to the Author):

Membrane bound O-acyltransferase (MBOAT) MBOAT7 enzyme plays important role in lipid metabolism. It also plays important role in controlling membrane lipid composition. The current manuscript provides a careful and detailed analysis of the biochemical and structural analysis of MBOAT7. The results are compelling. I have a few questions. They need to be carefully addressed in the revised manuscript.

We thank the reviewer for their critical and helpful evaluation of our manuscript. In response to the reviewer's critique, we revised our manuscript as detailed below.

1. The site for binding the substrate LPI was not elucidated. Please carefully address this shortcoming. Please indicate what needs to be done in order to solve the problem.

We agree that this is an important issue. We tried to acquire a structure of the complex of MBOAT7 and substrates (Extended Data Fig. 10). Unfortunately, and despite many attempts, the complex structure did not reach sufficient resolution to confidently assign density to substrates.

Because identifying the LPI in the structure was currently not feasible experimentally, we used a combination of computational and biochemistry approaches to elucidate how MBOAT7 binds lyso-PI. In the original manuscript, we modelled the interaction of MBOAT7 with lyso-PI by using published data on lyso-PC binding to chicken MBOAT5 structure (PDB: 7F3X). We observe similar structural features in MBOAT7 and MBOAT5. Consistent with their function in substrate binding, these residues are required for catalysis.

Also, we observed conformational changes of TM6 upon substrate binding from our lower resolution structure and designed experiments to test whether this conformational change is required for the optimal catalysis of the enzyme. Figure 3g shows that mutating residues in TM6 indeed interferes with efficient catalysis.

For the unique inositol headgroup of lyso-PI, we identified a series of conserved residues that are unique to MBOAT7 and are predicted to mediate substrate specificity by forming hydrogen bonds with the inositol group of lyso-PI. These inositol recognition residues enrich on a region that is variable among different lyso-phospholipid acylating MBOATs (MBOATs 1, 2, 5 and 7). Consistent with our hypothesis, this region confers substrate selectivity towards different headgroups among the 4 MBOATs – region swapping will exchange their substrate preference towards the different lyso-phospholipids – thus providing another line of evidence to support the proposed interaction mode between MBOAT7 and lyso-PI.

Beyond this, we added new information on MD simulations of lyso-PI bound MBOAT7 in a membrane. We find that the proposed binding mode of lyso-PI in MBOAT7 is stable with average RMSDs for protein and lyso-PI are ~3.5 and 7 Å during 500-ns simulations. Specifically, lyso-PI did not leave the binding pocket in any of our simulations. Also, from the analysis of the MD simulation trajectories, we identified two transiently stable conformations of lyso-PI when bound to MBOAT7, suggesting other alternative stable complex conformations may exist.

2. The authors look for a small molecule inhibitor of MBOAT7. MBOAT7 plays a key role in remodeling the FA composition of multiple phospholipids. The rationale for finding small molecule inhibitors is not clear to this reviewer. Please indicate why inhibiting MBOAT7 is a good thing. Why

not look for activators of MBOAT7? Is the MBOAT7 gene known to be regulated by SREBP or LXR? Please briefly discuss.

The rationale behind identifying MBOAT7 inhibitors is primarily based on the observation that high levels of MBOAT7 expression correlates with severity of progression of several human cancers, e.g., in particular clear cell renal carcinoma (ccRCC). Neumann et al., 2020 have shown in cell and xenograft models that attenuating MBOAT7 activity arrests cell cycle and halt cell proliferation in ccRCC. Therefore, we propose that MBOAT7 inhibitors may be therapeutically useful. We have added more information to clarify this rationale in the Introduction and Discussion sections of the revised manuscript.

As far as we know, MBOAT7 expression has not been shown to be directly regulated by SREBP nor LXR. However, there is a relationship, as the reviewer points out. Deletion or reduced expression of *MBOAT7* can somehow trigger the activation of SREBP-1c and de novo lipogenesis through a so far unknown mechanism (Xia et al., PMID:32859645). We have better clarified this again in the Introduction.

3. *The cryo-EM of MBOAT7 has been published last year. Please indicate what new information the current manuscript offers.*

To our knowledge, our bioRxiv paper published in 2021 was the first report of human MBOAT7 structure, and perhaps our preprint is what the reviewer is referring to. The structure for a related MBOAT, chicken MBOAT5, was published in 2021 (Zhang et al., PMID: 34824256). MBOAT5 catalyzes a different reaction – the acylation of LPC with primarily arachidonyl CoA, and as stated in our manuscript, we utilized insights from the MBOAT5 structure to work out mechanisms for MBOAT7. Additionally, we go well beyond the MBOAT5 paper content by a) identifying the specificity for different lyso-PL headgroups for different MBOATs and by b) identifying promising lead compounds for MBOAT7 inhibitors.

4. *The authors describe the biochemical enzyme assay for MBOAT7. The assay time is 7 minutes. What is the rationale? Please provide the ref for this assay. Is the enzyme stable in the assay condition described?*

We are not able to identify the “7 minute” assay this reviewer referred to. We used two assays to examine the activity of purified MBOAT7 and related mutants: a 3-minute fluorescence assay based on Nieng Yang’s and Xiaochun Li’s papers in 2020 (Wang et al., PMID: 32433610; Long et al., PMID: 32433613); and a 15-min radioactivity assay based our own studies (Sui, et al., PMID: 32433611). These two assays have been commonly used to assess acyltransferase activities of MBOAT enzymes including DGAT1, ACAT1, ACAT2 and MBOAT5.

To experimentally examine whether MBOAT7 protein is stable during a 15-min reaction at 37 °C, we conducted an experiment of first incubating MBOAT7 protein without substrates at 37 °C for 15 minutes, then performing the normal radioactivity assay by adding substrates and incubating at 37 °C for another 15 minutes. The measured activity is not significantly different from that of the fresh MBOAT7 without the pre-incubation phase (please see data below). We conclude that the purified MBOAT7 prep is stable through our activity assays. This finding is now mentioned in the Methods section as data not shown.

5. *ATR-101 is a potent ACAT1 inhibitor. It is clinically approved to treat a rare form of cancer in the adrenals. Here the authors write ATR101 as a prototypical ACAT1 inhibitor. Please note that ATR101 produces multiple off-target side effects as reported by Langlois et al published at PMID: PMC5932779. DOI: 10.1186/s12902-018-0251-5. Other ACAT1 inhibitor does not possess the same phenotypes as ATR101.*

Thank you for pointing out the off-target side effects of ATR101 drug. Indeed, we found that ATR101 showed significant inhibition of MBOAT7 using our activity assay. We have included this information and citations in our revised manuscript now.

Reviewer #2 (Remarks to the Author):

The focus of this manuscript is the membrane-bound O-acyltransferase 7, MBOAT7, responsible for acetylating lyso-PI with arachidonyl-CoA. In this work, Wang et al. present the structure of MBOAT7 determined by cryo-EM, alongside new mechanistic insights and determinants of phospholipid headgroup selectivity. MBOAT7 is a therapeutic target and an in silico screen presented here resulted in the identification of small-molecule inhibitors of MBOAT7.

Key findings of this work are that

- i) The TM6 of MBOAT7 was found to be a discontinuous helix, broken by Pro residues, the mutation of which led to reduced lyso-PI acylation activity. A lower resolution substrate-bound MBOAT7 cryoEM structure was used to demonstrate a shift in TM6, providing more support for a mechanistic role of the discontinuous TM6 helix.*
- ii) Loop-swap mutations of the TM4-TM5 region allowed the phospholipid selectivity of MBOATs to be altered, providing new insight into the recognition of lipid headgroups*
- iii) Molecular simulations were used to characterise MBOAT7-lyso-PI interactions, where the sn-1 acyl chain was observed to be extended into a hydrophobic channel*
- iv) A structure-based screen led to two compounds that were found to inhibit MBOAT7 activity*

Overall, this work represents a significant advance in our understanding of MBOATs involved in the Lands cycle and is valuable for our understanding of lipid specificity across other systems.

We thank the reviewer for their critical and helpful evaluation of our manuscript. We have revised the manuscript in response.

This reviewer has two main concerns to be addressed:

a. *The model used for the basis of molecular simulations has not been well-described. How was lyso-PI modelled into the cavity? It is not possible to judge the simulation quality as the only data presented from the simulation is an ‘energy-minimised structure’, it is not clear what this means in this context. Is this a representative conformation from the independent 500 ns simulations? It would be more valuable for the reader if more information was included, such as how the initial model was built and snapshots of the initial and final orientations of lyso-PI. It would also be of interest to see if the TM6 conformation changed in the presence or absence of lyso-PI during simulations. It was not described in the Methods how MBOAT7 was modelled into the POPC membrane.*

We thank the reviewer for pointing this out. We have now added detailed information on the simulations to the revised manuscript. In short, lyso-PI was inserted into MBOAT7 in a binding pose analogous to lyso-PC bound to chicken MBOAT5 (PDB 7F3X). We then put the complex structure in a POPC membrane in a solvated system using algorithms now described in the method section and the “orientations of proteins in membranes” (OPM) database as a reference. One round of energy minimization resulted in the energy-minimized structure. We compared this energy-minimized complex structure to the chicken MBOAT5 structure with lyso-PC in Extended Data Fig. 6.

We added a new figure and edited the result section for the MD simulation results. In the Extended Data Fig. 7, we show RMSDs of MBOAT7 and lyso-PI in trajectories of three independent 500-ns simulations. We also provided two representative snapshots of lyso-PI conformations after simulation (Extended Data Fig. 7 for comparison with the initial state presented in Fig.3).

We also performed three 500-ns simulations for membrane-embedded MBOAT7 without substrates. In all these simulations, we did not observe significant conformational changes of TM6. Please see below for the RMSF analysis of the alpha carbons over the trajectories. The most flexible regions are, as expected, the solvent-exposing loops that connect the TMs. TM6 is similarly stable to its neighbor TM5 and TM7. We were not able to observe the conformational changes of TM6 during these simulations possibly due to the short timescale of these simulations or due to the absence of acyl-CoA, as the conformational changes were only observed in the cryo-EM structure with two substrates. These simulation data of the apo MBOAT7 is made for the review purpose only and is not added to the revised manuscript.

b. *There is very little information on the VirtualFlow screening. More detail is needed on how the 12 candidate compounds were selected. How were scaffold similarity criteria incorporated into this decision?*

We thank the reviewer for pointing this out. We have now added more information for the VirtualFlow screening in the Methods section.

We selected the 12 candidates for further testing based on docking scores, fingerprint similarity, and substructures of top 50 compounds. We now added a new figure to show the ranking, hierarchical clustering of the fingerprints of the top 50 compounds, and classifications based on substructures in a newly added Extended Data Fig. 11.

This manuscript is very well presented and there are few minor edits:

c. *Extended Data Fig 4 caption, “corresponding residues are labelled in pair” unclear meaning, could be rephrased for clarity*

Thank you for pointing out this ambiguity. We have rephrased the sentence in the revised Extended Data Fig. 5 (previously Extended Data Fig. 6).

d. *Extended Data Fig 5 what lyso-PI structure? Is this a representative model from the molecular dynamics simulations presented here?*

We have clarified this in the legend. It is the same energy-minimized structure as shown in Figure 3a-d.

e. *Discussion of the possible basis of disease-causing mutations E353K and R384Q based on this new structure – this could be clearer for the reader if panels showing this were incorporated into Extended Data Fig. 9.*

Thank you for this suggestion. We have now added two extra panels in Extended Data Fig. 12 (originally Extended Data Fig.9). We showed the position and interaction with neighbor loop residues of E353 in Extended Data Figure 12b, and the relative position of R384 to the acyl-CoA binding tunnel in Extended Data Figure 12c. We have also added more explanation of these two mutations in the Discussion section in the revised manuscript.

Reviewer #3 (Remarks to the Author):

The manuscript under consideration presents the structure of the membrane-bound Oacyltransferase 7 (MBOAT7), which catalyzes the acylation of lyso-phosphatidylinositol (lyso-PI) with arachidonyl-CoA. The structure was obtained with single-particle cryoEM of amphipol-embedded (PMAL-C8) human MBOAT7 produced in HEK293 cells, leading to a reconstruction of 3.7Å global resolution. The authors use a fluorescence-based assay and a TLC assay to characterize the substrate specificity of the enzyme towards different donors and acceptors for the catalyzed acyl chain transfer reaction.

The authors were unable to obtain high-resolution structures of substrate bound MBOAT7, thus they proceeded to approximate the substrate-bound states using computational methods. In the case of arachidonyl-CoA (donor) they superimpose the MBOAT7 structure with that of cMBOAT5 bound to arachidonyl-CoA. For, Lyso-PI, the authors obtain a low resolution reconstruction to identify conformational changes within the transmembrane domain of MBOAT7, and additionally, they perform molecular dynamics simulations to characterize how the acceptor substrate fits within the

acceptor cavity. Importantly, in both cases the authors pair their computational predictions with mutagenesis experiments that appear to confirm the substrate binding predictions.

The workflows used for cryoEM structure determination are presented clearly, and have sufficient detail. The statistics presented for the structure are appropriate for the resolution ranges being reported. Importantly, the densities presented on extended data fig. 2 are consistent with the reported resolution. The presentation of a local resolution map and an angular distribution plot for the dataset was appreciated.

Overall, the manuscript represents high-quality and medium to high-impact work that advances our understanding of phospholipid biogenesis and modification. The manuscript is well-written and the methods section is sufficiently detailed to allow replication.

We thank the reviewer for their critical and helpful evaluation of our manuscript. In response to the reviewer's critique, we have revised our manuscript.

Some points for improvement are listed below:

1) *Line 149: It is mentioned that mutation of Met349 to Ala leads to reduced acylation activity, yet this residue is not present in the bar graph shown in Fig. 2d*

We apologized for this typo. Actually, we performed mutagenesis studies for the other 3 of the 4 mentioned residues that are in close contacts with the pantetheine group of arachidonyl-CoA, but not Met349. We have corrected this in the revised manuscript.

2) *In Extended fig 8 the two states of the protein (apo and lyso-PI – bound) are shown at low resolution for comparison. In text (line 196), it is mentioned that the lateral hydrophobic window is enlarged in the lyso-PI-bound state. Adding an alternative view like in fig 3c, for apo vs bound, would highlight better the purported difference.*

We thank the reviewer for this suggestion. The hydrophobicity surface representations showed in Fig. 3 are from the MBOAT7 apo model. As we currently only have a lower resolution structure of MBOAT7 bound to lyso-PI, we were not able to build a model for this. We hope this reviewer can understand that it is difficult to generate this parallel comparison panels for now.

To better visualize the differences in the EM maps we now added circles to highlight the openings of the lateral channels.

3) *The authors have used the Alphafold2 predicted model as an initial model for constructing their model based on the cryoEM reconstruction. I believe that a comparison between the starting (Alphafold2) model and the final (cryoEM) model would be of great interest to readers as a test case.*

Thank you for the suggestion. We agree with the reviewer that this comparison can be informative and have added these data in Extended Data Figure 3. We also included the RMSDs among different comparisons.

4) *On Fig. 2c a visual grouping (something like a colored bracket) of the residues that contribute to the fatty acid binding pocket could help the reader to better orient themselves within the presented active site (correspondence between Fig. 2b and 2c)*

Thank you for this helpful suggestion. We have revised Figure 2 accordingly.

5) For the model validation a MolProbity score and Clashscore is provided. It would be nice to also see some map to model type validation like the Q score that is included in the PDB validation report or an EM ringer score. A description of the score(s) used should be included in the methods section.

Finally, a 3DFSC sphericity score should be included for quality assessment of the data.

Thank you for the suggestion. We have added the Q score and 3D FSC sphericity score to the EM data report, and the related description and citations in the method section.

Reviewer #4 (Remarks to the Author):

The paper “The structure, catalytic mechanism, and inhibitor identification of phosphatidylinositol remodelling MBOAT7” describes novel structure of MBOAT7. The protein functionality has been confirmed by biochemical and biophysical methods, mutagenesis and chimeric-proteins modifications. Very detailed analysis of the structure and substrate docking allowed to suggest mechanics of possible substrate binding and its specificity. Authors speculate on the effect of disease related mutations based on their position on the structure. Paper is well written and easy to follow, conclusions are justified, experimental methods have enough details to reproduce the methodology. The paper contains high quality illustrations and detailed methodology. I recommend publication after minor corrections.

We thank the reviewer for their critical and helpful evaluation of our manuscript. In response to the reviewer’s critique, our manuscript has undergone revisions.

Comments:

It is known that loss-of-function mutations in human MBOAT7 results in liver disease, intellectual disability, early onset seizures, and autism spectrum disorders” Could use in Inhibitors like Sevenin1 and -2. Use of Inhibitors will be equivalent to the “loss of function”. Will protein inhibition cause similar diseases as mutations with the loss of function? Silencing the gene also seems lethal. I would like authors to add extra thoughts on benefits of inhibitors and how to avoid mutation-like effects by using inhibitors.

Thank you for the question. We developed the inhibitors as reagents to treat primarily MBOAT7-dependent cancers. As this reviewer stated, loss-of-function of MBOAT7 can lead to early brain developmental defects, and eventually juvenile lethality. However, genetic alleles that reduced the expression of *MBOAT7* are viable, and cause fatty liver diseases. We think the inhibitors can still be useful since aggressive cancers (e.g., ccRCC) would require a shorter time treatment during which development of fatty liver disease is less of an issue and because Sevenins (or related compounds) could be designed not to cross the brain blood barrier (BBB). We edited the discussion section to better explain this.

Minor comments

1. please check if you refer to correct figure

Thanks for the suggestion. We have gone through the manuscript and made sure the text and figures match in the revised manuscript.

Line 122 “multiple conformations in the apo state of the enzyme (Extended Data Fig. 2f).” Figure 2f shows density with fitted helices, please refer to correct figure.

Thanks for pointing this out. This actually refers to the correct figure. The density shown for the transmembrane helices are contoured at 6σ , except TM8 is shown at two contour levels. This is because TM8 is more flexible than other helices and we observed a weaker density for TM8 than other TM helices. We needed to use lower contour level (4σ) to be able to build the model for it. We have edited the manuscript to clarify.

REVIEWERS' COMMENTS

Reviewer #1 (Remarks to the Author):

Acceptable

Reviewer #2 (Remarks to the Author):

The revisions made to the manuscript address the previous concerns.